# Parsing for Universal Dependencies without training

## Abstract

We present UDP, an unsupervised parsing algorithm for Universal Dependencies (UD) based on PageRank and a small set of specific dependency head rules. The parser requires no training, and it is competitive with a delexicalized transfer system. UDP offers a linguistically sound unsupervised alternative to cross-lingual UD parsing. It is distinctly robust to domain change across languages.

## 1 Introduction

Grammar induction and unsupervised dependency parsing are active fields of research in natural language processing (Klein and Manning, 2004; Gelling et al., 2012). However, many data-driven approaches struggle with learning relations that match the conventions of the test data, e.g., Klein and Manning reported the tendency of their DMV parser to make determiners the heads of German nouns. Even supervised transfer approaches (McDonald et al., 2011) suffer from target adaptation problems when facing word order differences.

The Universal Dependencies (UD) project (Nivre et al., 2015) offers a dependency formalism that aims at providing a consistent representation across languages, while enforcing a few hard constraints. The arrival of such treebanks, expanded and improved on a regular basis, provides a new milestone for cross-lingual dependency parsing research (McDonald et al., 2013). Furthermore, we expect that such a formalism lends itself more naturally to a simple and linguistically sound rule-based approach to cross-lingual parsing. In this paper we present such an approach.

Our system is a dependency parser that requires no training, and relies solely on explicit part-of-speech (POS) constraints that UD imposes. In particular, UD prescribes that trees are single-rooted, and that function words like adpositions, auxiliaries, and determiners are always dependents of content words, while other formalisms might treat them as heads (De Marneffe et al., 2014).

**Contributions**  Our method goes beyond the existing work on *rule-aided* unsupervised dependency parsing by a) adapting dependency head rules to UD-compliant POS relations, b) incorporating the UD restriction of function words being leaves, c) using personalized PageRank to improve main predicate identification, and d) making it completely free of language-specific parameters by estimating adposition attachment direction directly on test data.

We evaluate our system on 32 languages[1] in three setups, depending on the reliability of available POS tags, and compare to a multi-source delexicalized transfer system. In addition, we evaluate the systems' sensitivity to domain change for a subset of UD languages for which domain information was retrievable. The results expose a solid and competitive system for all UD languages. Our unsupervised parser compares favorably to delexicalized parsing, while being more robust to domain change across languages.

## 2 Related work

Over the recent years, cross-lingual linguistic structure prediction based on model transfer or projection of POS tags and dependency trees has become a relevant line of work (Das and Petrov, 2011; McDonald et al., 2011). These works mostly use supervised learning and different target language adaptation techniques.

---

[1] Out of 33 languages in UD v1.2. We exclude Japanese because the treebank is distributed without word forms and hence we can not provide results on predicted POS.

The first group of approaches deals with annotation projection (Yarowsky et al., 2001), whereby parallel corpora are used to transfer annotations between resource-rich source languages and low-resource target languages. Projection relies on the availability and quality of parallel corpora, source-side taggers and parsers, but also tokenizers, sentence aligners, and word aligners for sources and targets. Hwa et al. (2005) were the first to project syntactic dependencies, and Tiedemann (2014) improved on their projection algorithm. Current state of the art in cross-lingual dependency parsing involves leveraging parallel corpora (Ma and Xia, 2014; Rasooli and Collins, 2015).

The second group of approaches deals with transferring source parsing models to target languages. Zeman and Resnik (2008) were the first to introduce the idea of delexicalization: removing lexical features by training and cross-lingually applying parsers solely on POS sequences. Søgaard (2011) and McDonald et al. (2011) independently extend delexicalization to involve multiple source-side parsers. This line of work depends on applying uniform POS and dependency representations (McDonald et al., 2013).

Both model transfer and annotation projection rely on a large number of presumptions to derive their competitive parsing models. By and large, these presumptions are unrealistic and exclusive to a group of very closely related, resource-rich Indo-European languages. Agić et al. (2015) expose some of the biases in their proposal for realistic cross-lingual POS tagging, as they emphasize the lack of perfect sentence and word splitting for truly low-resource languages. Johannsen et al. (2016) introduce joint projection of POS and dependencies from multiple sources while sharing the outlook on bias removal in real-world multilingual processing.

Cross-lingual learning, realistic or not, depends entirely on the availability of data: for the sources, for the targets, or most often for both sets of languages. Moreover, it typically does not exploit the constraints placed on the linguistic structures through the formalism, and it does so by design. With the emergence of UD as the practical standard for cross-language annotation of POS and syntactic dependencies, we argue for an approach that takes a fresh angle on both aspects. Namely, we propose a parser that i) requires *no* training data, and in contrast ii) critically relies on exploiting the UD constraints on building POS and dependency annotations.

These two characteristics make our parser unsupervised. Data-driven unsupervised dependency parsing is a well-established discipline (Klein and Manning, 2004; Spitkovsky et al., 2010a; Spitkovsky et al., 2010b). Still, the performance of unsupervised parsers falls far behind the approaches involving any sort of supervision.

Our work builds on the research on rule-aided unsupervised dependency parsing (Gillenwater et al., 2010; Naseem et al., 2010; Søgaard, 2012a; Søgaard, 2012b). In particular, we make use of Søgaard's (2012b) PageRank method to rank words before decoding. Our system, however, has two key differences: i) the usage of PageRank personalization, and of ii) two-step decoding to treat content and function words differently according to the UD formalism. Through these differences, even without any training data, we parse nearly as well as a delexicalized transfer parser, and with increased stability to domain change.

## 3 Method

Our approach does not use any training or unlabeled data. We have used the English treebank during development to assess the contribution of individual head rules, and to tune PageRank parameters (Sec. 3.1) and function-word directionality (Sec. 3.2). Adposition direction is calculated on the fly on test data. In the following, we refer to our UD parser as UDP.

### 3.1 PageRank setup

Our system uses the PageRank (PR) algorithm (Page et al., 1999) to estimate the relevance of the content words of a sentence. PR gives higher rank to nodes with more incoming edges, as well as to nodes connected to those. Using PR to score word relevance requires an effective graph-building strategy. We have experimented with the strategies by Søgaard (2012b), but our system fares best strictly using the dependency rules in Table 1 to build the graph.

We build a multigraph of all words in the sentence covered by the head-dependent rules in Table 1, giving each word an incoming edge for each eligible dependent, i.e., ADV depends on ADJ and VERB. This strategy does not always yield connected graphs, and we use a teleport probability of 0.05 to ensure PR convergence. We chose this

value incrementally in intervals of 0.01 during development until we found the smallest value that guaranteed PR convergence. A high teleport probability is undesirable, because the resulting stationary distribution can be almost uniform. We did not have to re-adjust this value when running on the actual test data.

The the main idea behind our personalized PR approach is the observation that ranking is only relevant for content words.[2] PR can incorporate a priori knowledge of the relevance of nodes by means of *personalization*, namely giving more weight to certain nodes. Intuitively, the higher the rank of a word, the closer it should be to the root node, i.e., the main predicate of the sentence is the node that should have the highest PR, making it the dependent of the root node (Fig. 1, lines 4-5). We use PR personalization to give 5 times more weight (over an otherwise uniform distribution) to the node that is estimated to be main predicate, i.e., the first verb or the first content word if there are no verbs.

### 3.2 Head direction

Head direction is an important syntactic trait. Indeed, the UD feature inventory contains a trait to distinguish adposition between pre- and post-positions. Instead of relying on this feature from the treebanks, which is not always provided, we estimate the frequency of ADP-NOMINAL vs. NOMINAL-ADP bigrams.[3] This estimation requires very few examples to converge (10-15 sentences), and we calculate it directly on *test* data.

If a language has more ADP-NOMINAL bigrams, we consider all its ADP to be prepositions (and thus dependent of elements at their right). Otherwise, we consider them to be postpositions.

For other function words, we have determined on the English dev data whether to make them strictly right- or left-attaching, or to allow either direction: AUX, DET, and SCONJ are right-attaching, while CONJ and PUNCT are left-attaching. There are no direction constraints for the rest.

### 3.3 Decoding

Fig. 1 shows the tree-decoding algorithm. It has two blocks, namely a first block (3-11) where we assign the head of content words according to

---

[2]ADJ, NOUN, PROPN, and VERB mark content words.

[3]NOMINAL= {NOUN, PROPN, PRON}

1: $H = \emptyset; D = \emptyset$
2: $C = \langle c_1, ... c_m \rangle; F = \langle f_1, ... f_m \rangle$
3: **for** $c \in C$ **do**
4: **if** $|H| = 0$ **then**
5: $h = root$
6: **else**
7: $h = \text{argmin}_{j \in H} \{\gamma(j, c) \mid \delta(j, c) \wedge \kappa(j, c)\}$
8: **end if**
9: $H = H \cup \{c\}$
10: $D = D \cup \{(h, c)\}$
11: **end for**
12: **for** $f \in F$ **do**
13: $h = \text{argmin}_{j \in H} \{\gamma(j, f) \mid \delta(j, f) \wedge \kappa(j, f)\}$
14: $D = D \cup \{(h, f)\}$
15: **end for**
16: **return** $D$

Figure 1: Two-step decoding algorithm for UDP.

---

| ADJ $\longrightarrow$ ADV |
| VERB $\longrightarrow$ ADV, AUX, NOUN, PROPN, PRON, SCONJ |
| NOUN, PROPN $\longrightarrow$ ADP, DET, NUM |
| NOUN, PROPN $\longrightarrow$ ADJ, NOUN, PROPN |

Table 1: UD dependency rules

their PageRank and the constraints of the dependency rules, and a second block (12-15) where we assign the head of function words according to their proximity, direction of attachment, and dependency rules. The algorithm requires:

1. The PR-sorted list of content words $C$.
2. The set of function words $F$.
3. A set $H$ for the current possible heads, and a set $D$ for the dependencies assigned at each iteration, which we represent as head-dependent tuples $(h, d)$.
4. A symbol $root$ for the root node.
5. A function $\gamma(n, m)$ that gives the linear distance between two nodes.
6. A function $\kappa(h, d)$ that returns whether the dependency $(h, d)$ has a valid attachment direction given the POS of the $d$ (cf. Sec. 3.2).
7. A function $\delta(h, d)$ that determines whether $(h, d)$ is licensed by the rules in Table 1.

The head assignations in lines 7 and 13 read as follow: the head $h$ of a word (either $c$ or $f$) is the closest element of the current list of heads ($H$) that has the right direction ($\kappa$) and respects the POS-dependency rules ($\delta$). These assignations have a back-off option to ensure the final $D$ is a tree. If the conditions determined by $\kappa$ and $\delta$ are too strict,

i.e. if the set of possible heads is empty, we drop the $\delta$ head-rule constraint and recalculate the closest possible head that respects the directionality imposed by $\kappa$. If the set is empty again, we drop both constraints and assign the closest head.

Lines 4 and 5 enforce the single-root constraint. To enforce the leaf status of function nodes, the algorithm first attaches all content words ($C$), and then all function words ($F$) in the second block where H is not updated, thereby ensuring leafness for all $f \in F$. The order of head attachment is not monotonic wrt. PR between the first and second block, and can yield non-projectivities. Nevertheless, it still is a one-pass algorithm. Decoding runs in less than $O(n^2)$, namely $O(n \times |C|)$. However, running PR incurs the main computation cost.

## 4 Parser run example

This section exemplifies a full run of UDP for the example sentence "They also had a special connection to some extremists", an actual clause from the English test data.

### 4.1 PageRank

Given an input sentence and its POS tags, we obtain rank of each word by building a graph using head rules and running PR on it. Table 2 provides the sentence, the POS of each word, the number of incoming edges for each word after building the graph with the head rules from Sec. 3.1, and the personalization vector for PR on this sentence. Note that all nodes have the same personalization weight, except the estimated main predicate, the verb "had".

| | Word: | They | also | had | a | special | connection | to | some | extremists |
|---|---|---|---|---|---|---|---|---|---|---|
| | POS: | PRON | ADV | VERB | DET | ADJ | NOUN | ADP | DET | NOUN |
| Personalization: | | 1 | 1 | 5 | 1 | 1 | 1 | 1 | 1 | 1 |
| Incoming edges: | | 0 | 0 | 4 | 0 | 1 | 5 | 0 | 0 | 5 |

Table 2: Words, POS, Personalization and incoming edges for the example sentence.

Table 3 shows the directed multigraph used for PR in detail. We can see e.g. that the four incoming edges for the verb "come" from the two nouns, plus from the adverb "also" and the pronoun "They".

After running PR, we obtain the following ranking for content words:
$C = \langle$had,connection,extremists,special$\rangle$
Even though the verb has four incoming edges and the nouns have six each, the personalization makes the verb the highest-ranked word.

| → | They | also | had | a | special | connection | to | some | extremists |
|---|---|---|---|---|---|---|---|---|---|
| They | - | | • | | | • | | | |
| also | | - | • | | • | | | | |
| had | | | - | | | | | | |
| a | | | | - | | • | | | • |
| special | | | | | - | • | | | • |
| connection | | | • | | | - | | | • |
| to | | | | | | • | - | | • |
| some | | | | | | • | | - | • |
| extremists | | | • | | | • | | | - |

Table 3: Matrix representation of the directed graph for the words in the sentence.

### 4.2 Decoding

Once $C$ is calculated, we can follow the algorithm in Fig. 1 to obtain a dependency parse. Table 4 shows a trace of the algorithm, given $C$ and $F$:
$C = \langle$had,connection,extremists,special$\rangle$
$F = \{$They, also, a, to, some$\}$

The first four iterations calculate the head of content words following their PR, and the following iterations attach the function words in $F$.

| it | word | h | H |
|---|---|---|---|
| 1 | had | root | ∅ |
| 2 | connection | had | {had} |
| 3 | extremists | had | {had, connection} |
| 4 | special | connection | {had, connection, extremists} |
| 5 | They | had | {had, connection, extremists, special} |
| 6 | also | had | ... |
| 7 | a | connection | ... |
| 8 | to | extremists | ... |
| 9 | some | extremists | ... |

Table 4: Algorithm trace for example sentence. $It =$iteration number, $word =$current word, $H =$ current set of possible heads.

Finally, Fig. 2 shows the resulting dependency tree. Full lines are assigned in the first block (content dependents), dotted lines are assigned in the second block (function dependents). The edge labels indicate in which iteration the algorithm has assigned each dependency.

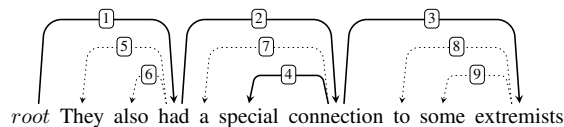
$root$ They also had a special connection to some extremists

Figure 2: Example dependency tree predicted by the algorithm.

Note that the algorithm is deterministic for a certain input POS sequence. Any 10-token sentence with the POS labels shown in Table 2 would yield the same dependency tree.[4]

---

[4]The resulting trees always pass the validation script in

| Language | $BL_G$ | $UDP_G$ | $MSD_G$ | $MSD_P$ | $UDP_P$ | $UDP_N$ |
|---|---|---|---|---|---|---|
| Ancient Greek | 42.2 L | 43.4 | 48.6 | 46.5 | 41.6 | 27.0 |
| Arabic | 34.8 R | 47.8 | 52.8 | 52.6 | 47.6 | 41.0 |
| Basque | 47.8 R | 45.0 | 51.2 | 49.3 | 43.1 | 22.8 |
| Bulgarian | 54.9 R | 70.5 | 78.7 | 76.6 | 68.1 | 27.1 |
| Church Slavonic | 53.8 L | 59.2 | 61.8 | 59.8 | 59.2 | 35.2 |
| Croatian | 41.6 L | 56.7 | 69.1 | 65.6 | 54.5 | 25.2 |
| Czech | 46.5 R | 61.0 | 69.5 | 67.6 | 59.3 | 25.3 |
| Danish | 47.3 R | 57.9 | 70.2 | 65.6 | 53.8 | 26.9 |
| Dutch | 36.1 L | 49.5 | 57.0 | 59.2 | 50.0 | 24.1 |
| English | 46.2 R | 53.0 | 62.1 | 59.9 | 51.4 | 27.9 |
| Estonian | 73.2 R | 70.0 | 73.4 | 66.1 | 65.0 | 25.3 |
| Finnish | 43.8 R | 45.1 | 52.9 | 50.4 | 43.1 | 21.6 |
| French | 47.1 R | 64.5 | 72.7 | 70.6 | 62.1 | 36.3 |
| German | 48.2 R | 60.6 | 66.9 | 62.5 | 57.0 | 24.2 |
| Gothic | 50.2 L | 57.5 | 61.7 | 59.2 | 55.8 | 34.1 |
| Greek | 45.7 R | 58.5 | 68.0 | 66.4 | 57.0 | 29.3 |
| Hebrew | 41.8 R | 55.4 | 62.0 | 58.6 | 52.8 | 35.7 |
| Hindi | 43.9 R | **46.3** | 34.6 | 34.5 | **45.7** | 27.0 |
| Hungarian | 53.1 R | 56.7 | 58.4 | 56.8 | 54.8 | 22.7 |
| Indonesian | 44.6 L | 60.6 | 63.6 | 61.0 | 58.4 | 35.3 |
| Irish | 47.5 R | 56.6 | 62.5 | 61.3 | 53.9 | 35.8 |
| Italian | 50.6 R | 69.4 | 77.1 | 75.2 | 67.9 | 37.6 |
| Latin | 49.4 L | 56.2 | 59.8 | 54.9 | 52.4 | 37.1 |
| Norwegian | 49.1 R | 61.7 | 70.8 | 67.3 | 58.6 | 29.8 |
| Persian | 37.8 L | 55.7 | 57.8 | 55.6 | 53.6 | 33.9 |
| Polish | 60.8 R | 68.4 | 75.6 | 71.7 | 65.7 | 34.6 |
| Portuguese | 45.8 R | 65.7 | 72.8 | 71.4 | 64.9 | 33.5 |
| Romanian | 52.7 R | 63.7 | 69.2 | 64.0 | 58.9 | 32.1 |
| Slovene | 50.6 R | 63.6 | 74.7 | 71.0 | 56.0 | 24.3 |
| Spanish | 48.2 R | 63.9 | 72.9 | 70.7 | 62.1 | 35.0 |
| Swedish | 52.4 R | 62.8 | 72.2 | 67.2 | 58.5 | 25.3 |
| Tamil | 41.4 R | 34.2 | 44.2 | 39.5 | 32.1 | 20.3 |
| *Average* | 47.8 | 57.5 | 63.9 | 61.2 | 55.3 | 29.9 |

Table 5: UAS for baseline with gold POS ($BL_G$) with direction (L/R) for backoff attachments, UDP with gold POS ($UDP_G$) and predicted POS ($UDP_P$), PR with naive content-function POS ($UDP_N$), and multi-source delexicalized with gold and predicted POS ($MSD_G$ and $MSD_P$, respectively). BL values higher than $UDP_G$ are underlined, and $UDP_G$ values higher than $MSD_G$ are in boldface.

# 5 Experiments

This section describes the data and metrics used to assess the performance of UDP, as well as the systems we compare against. We evaluate on the test sections of the UD1.2 treebanks (Nivre et al., 2015) that contain word forms. If there is more than one treebank per language, we use the treebank that has the canonical language name (e.g. *Finnish* instead of *Finnish-FTB*).

It is common to evaluate unsupervised dependency parsing using alternative metrics like undirected attachment score or neutralized edge direction, or to evaluate short sentences separately (Schwartz et al., 2011; Gelling et al., 2012). In contrast, we use standard unlabeled attachment score (UAS) and evaluate on all sentences of the

github.com/UniversalDependencies/tools

canonical UD test sets.

## 5.1 Baseline

We compare our UDP system with the performance of a rule-based baseline that uses the head rules in Table 5. The baseline identifies the first verb (or first content word if there are no verbs) as the main predicate, and assigns heads to all words according to the rules in Table 1.

We have selected the set of head rules to maximize precision on the development set, and they do not provide full coverage. The system makes any word not covered by the rules (e.g., a word with a POS such as X or SYM) either dependent of their left or right neighbor, depending on the estimated runtime parameter.

We report the best head direction and its score for each language in Table 5. This baseline finds the head of each token based on its closest possible head, or on its immediate left or right neighbor if there is no head rule for the POS at hand, which means that this system does not necessarily yield well-formed tress. Each token receives a head, and while the structures are single-rooted, they are not necessarily connected.

Note that we do not include results for the DMV model by Klein and Manning (2004), as it has been outperformed by a system similar to ours (Søgaard, 2012b). The usual adjacency baseline for unsupervised dependency parsing, where all words depend on their left or right neighbor, fares much worse than our baseline (20% UAS below on average) even when we make an oracle pick for the best per-language direction. Therefore we do not report those scores.

## 5.2 Evaluation setup

Our system relies solely on POS tags. To estimate the quality degradation of our system under non-gold POS scenarios, we evaluate UDP on two alternative scenarios. The first is predicted POS ($UDP_P$), where we tag the respective test set with TnT (Brants, 2000) trained on each language's training set. The second is a naive type-constrained two-POS tag scenario ($UDP_N$), and approximates a lower bound. We give each word either CONTENT or FUNCTION tag, depending on the word's frequency. Words that belong to the 100 most frequent word types of the input test section receive the FUNCTION tag.

Finally, we compare our system to a supervised cross-lingual system (MSD). It is a multi-

source delexicalized transfer parser, referred to as *multi-dir* in the original paper by McDonald et al. (2011). For this baseline we train Tur-boParser (Martins et al., 2013) on a delexicalized training set of 20k sentences, sampled uniformly from the UD training data excluding the target language. MSD is a competitive baseline in cross-lingual transfer parsing work. This gives us an indication how our system compares to standard cross-lingual parsers.

## 5.3 Results

Table 5 shows that UDP is a competitive system; because $UDP_G$ is remarkably close to the supervised $MSD_G$ system, with an average difference of 6.4%, even outperforming $MSD_G$ on one language (Hindi).

More interestingly, on the evaluation scenario with predicted POS we observe that our system drops only marginally (2.2%) compared to MSD (2.7%). In the least robust rule-based setup, the error propagation rate from POS to dependency would be doubled, as either a wrongly tagged head or dependent would break the dependency rules. However, with an average POS accuracy by TnT of 94.1%, the error propagation is 0.37, i.e each POS error causes 0.37 additional dependency errors. In contrast, for the MSD system this error propagation is 0.46, thus higher.[5]

For the extreme POS scenario, content vs. function POS (CF), the drop in performance for UDP is however very large. But this might be a too crude evaluation setup. Nevertheless, UDP, the simple unsupervised system with PageRank, outperforms the adjacency baselines (BL) by 4% on average on the two type-based naive POS tag scenario. This difference indicates that even with very deficient POS tags, UDP can provide better structures.

## 6 Discussion

In this section we provide a further error analysis of the UDP parser. We examine the contribution to the overal results of using PageRank to score content words, the behavior of the system across different parts of speech, and we assess the robustness of UDP when parsing text from different domains.

---

[5]Err. prop. $= (E(Parse_P) - E(Parse_G))/E(POS_P)$, where $E(x) = 1 - Accuracy(x)$.

## 6.1 PageRank contribution

The performance of UDP depends on PageRank to score content words, and on two-step decoding to ensure the leaf status of function words. In this section we isolate the constribution of both parts. We do so by comparing the performance of BL, UDP, and $UDP_{NoPR}$, a version of UDP where we disable PR and rank content words according to their reading order, i.e. the first word in the ranking is the first word to be read, regardless of the specific language's script direction

The baseline BL described in 5.1 already ensures function words are leaf nodes, because they have no listed dependent POS in the head rules. The task of the decoding steps is mainly to ensure the resulting structures are well-formed dependency trees.

However, if we measure the difference between $UDP_{NoPR}$ and BL, we observe that $UDP_{NoPR}$ contributes with 4 UAS points on average over the baseline. Nevertheless, the baseline is oracle-informed about the language's best branching direction, a property that UDP does not have. Instead, the decoding step determines head direction as described in Section 3.2.

Complementary, we can measure the contribution of PR by observing the difference between regular UDP and $UDP_{NoPR}$. The latter scores on average 9 UAS points lower than UDP. These 9 points are strictly determined by the better attachment of content words.

## 6.2 Breakdown by POS

UD is a constantly-improving effort, and not all v1.2 treebanks have the same level of formalism compliance. Thus, the interpretation of, e.g., the AUX-VERB or DET-PRON distinctions might differ. However, we do not incorporate these differences in our analysis and consider all treebanks equally compliant.

The root accuracy scores oscillate around an average of 69%, with Arabic and Tamil (26%) and Estonian (93%) as outliers. Given the PR personalization (Sec. 3.1), UDP has a strong bias for chosing the first verb as main predicate. However, without personalization, performance drops 2% on average. This improvement is consistent even for verb-final languages like Hindi. Moreover, our personalization strategy makes PR converge a whole order of magnitude faster.

The bigram heuristic to determine adposition

| Language | $BL_G$ | $MSD_G$ | $UDP_G$ | $MSD_P$ | $UDP_P$ |
|---|---|---|---|---|---|
| Bulgarian | 50.1±2.4 | 73.5±3.5 | 69.7±1.8 | 71.3±3.3 | 66.9±3.2 |
| Croatian+Serbian | 42.1±0.7 | 66.0±3.0 | 57.8±1.4 | 62.1±3.0 | 54.4±2.0 |
| English | 42.2±2.8 | 60.1±6.2 | 53.9±2.5 | 57.3±4.3 | 52.0±3.3 |
| Italian | 50.3±1.2 | 70.0±5.4 | 70.1±3.3 | 68.1±6.0 | 68.7±3.9 |
| *Average Std.* | 1.8 | 4.5 | 2.5 | 4.2 | 3.1 |

Table 6: Average language-wise domain evaluation. We report average UAS and standard deviation per language. The bottom row provides the average standard deviation for each system.

direction succeeds at identifying the predominant pre- or postposition preference for all languages (average ADP UAS of 75%). The fixed direction for the other functional POS is largely effective, with few exceptions, e.g., DET is consistently right-attaching on all treebanks except Basque (average overall DET UAS of 84%, 32% for Basque). These alternations could also be estimated from the data in a manner similar to ADP. Our rules do not make nouns eligible heads for verbs. As a result, the system cannot infer relative clauses. We have excluded the NOUN → VERB head rule during development because it makes the hierarchical relation between verbs and nouns less conclusive.

We have not excluded punctuation from the evaluation. Indeed, the UAS for the PUNCT is low (an average of 21%, standard deviation of 9.6), even lower than the otherwise problematic CONJ. Even though conjunctions are pervasive and identifying their scope is one of the usual challenges for parsers, the average UAS for CONJ is much larger (an average of 38%, standard deviation of 13.5) than for PUNCT. Both POS show large standard deviations, which indicates great variability. This variability can be caused by linguistic properties of the languages or evaluation datasets, but also by differences in annotation convention.

## 6.3 Cross-domain consistency

Models with fewer parameters are less likely to be overfit for a certain dataset. In our case, a system with few, general rules is less likely to make attachment decisions that are very particular of a certain language or dataset. Plank and van Noord (2010) have shown that rule-based parsers can be more stable to domain shift. We explore if their finding holds for UDP as well, by testing on i) the UD development data as a readily available proxy for domain shift, and ii) manually curated domain splits of select UD test sets.

| Language | Domain | $BL_G$ | $MSD_G$ | $UDP_G$ | $MSD_P$ | $UDP_P$ |
|---|---|---|---|---|---|---|
| Bulgarian | `bulletin` | 48.3 | _67.5_ | _67.4_ | 65.4 | _61.5_ |
| | `legal` | _47.9_ | 76.9 | 69.2 | 73.0 | 68.6 |
| | `literature` | 53.6 | 74.2 | 69.0 | 72.8 | 66.6 |
| | `news` | 49.3 | 74.6 | 70.2 | 73.0 | 68.2 |
| | `various` | 51.4 | 74.2 | 72.5 | 72.6 | 69.5 |
| Croatian | `news` | _41.2_ | _62.4_ | 57.9 | 61.8 | _52.2_ |
| | `wiki` | 41.9 | 64.8 | _55.8_ | _58.2_ | 56.3 |
| English | `answers` | 44.1 | 61.6 | 55.9 | 59.5 | 53.7 |
| | `email` | 42.8 | 58.8 | 52.1 | 57.1 | 56.3 |
| | `newsgroup` | _41.7_ | 55.5 | _49.7_ | 52.9 | 51.1 |
| | `reviews` | 47.4 | 66.8 | 54.9 | 63.9 | 52.2 |
| | `weblog` | 43.3 | _51.6_ | 50.9 | _49.8_ | 53.8 |
| | `magazine†` | 41.4 | 60.9 | 55.6 | 58.4 | 53.3 |
| | `bible†` | 38.4 | 56.2 | 56.2 | 56.8 | 48.6 |
| | `questions†` | 38.7 | 69.7 | 55.6 | 60.5 | _47.2_ |
| Italian | `europarl` | 50.8 | _64.1_ | **70.6** | _62.7_ | **69.7** |
| | `legal` | 51.1 | 67.9 | **69.0** | _64.4_ | **67.2** |
| | `news` | 49.4 | 68.9 | 67.5 | 67.0 | _65.3_ |
| | `questions` | _48.7_ | 80.0 | 77.0 | 79.1 | 76.1 |
| | `various` | 49.7 | 67.8 | **69.0** | 65.3 | **67.6** |
| | `wiki` | 51.8 | 71.2 | 68.1 | 70.3 | 66.6 |
| Serbian | `news` | 42.8 | _68.0_ | 58.8 | 65.6 | _53.3_ |
| | `wiki` | _42.4_ | 68.9 | 58.8 | 62.8 | 55.8 |

Table 7: Evaluation across domains. UAS for baseline with gold POS ($BL_G$), UDP with gold POS ($UDP_G$) and predicted POS ($UDP_P$), and multi-source delexicalized with gold and predicted POS ($MSD_G$ and $MSD_P$). English datasets marked with † are in-house annotated. Lowest results per language underlined. Bold: UDP outperforms MSD.

**Development sets** We have used the English development data to choose which relations would be included as head rules in the final system (Cf. Table 1). It would be possible that some of the rules are indeed more befitting for the English data or for that particular section.

However, if we regard the results for $UDP_G$ in Table 5, we can see that there are 24 languages (out of 32) for which the parser performs better than for English. This result indicates that the head rules are general enough to provide reasonable parses for languages other than the one chosen for development.

If we run $UDP_G$ on the development sections for the other languages, we find the results are very consistent. Any language scores on average ±1 UAS with regards to the test section. There is no clear tendency for either section being easier to parse with our system.

**Cross-domain test sets** To further assess the cross-domain robustness, we retrieved the domain (genre) splits[6] from the test sections of the UD

---

[6]The data splits are freely available at http://ANONYMIZED

treebanks where the domain information is available as sentence metadata: from Bulgarian, Croatian, and Italian. We also include a UD-compliant Serbian dataset which is not included in the UD release but which is based on the same parallel corpus as Croatian and has the same domain splits (Agić and Ljubešić, 2015). When averaging we pool Croatian and Serbian together as they come from the same dataset. Also, we use a tagger trained on the Croatian UD training data for tagging Serbian.

For English, we have obtained the data splits in the test section matching the sentences from the original distribution of the English Web Treebank. In addition to these already available data sets, we have annotated three different datasets to asses domain variation more extensively, namely the first 50 verses of the King James Bible, 50 sentences from a magazine, and 75 sentences from the test split in QuestionBank (Judge et al., 2006). We include the third dataset to evaluate strictly on questions, which we could do already in Italian. While the `answers` domain in English is made up of text from the Yahoo! Answers forum, only one fourth of the sentences are questions. Note these three small datasets are not included in the results on the canonical test sections in Table 5.[7]

Table 6 summarizes the per-language average score and standard deviation, as well as the macro-averaged standard deviation across languages. UDP has a much lower standard deviation across domains compared to MSD. This holds across languages. We attribute this higher stability to UDP being developed to satisfy a set of general properties of the UD syntactic formalism, instead of being a data-driven method more sensitive to sampling bias. This holds for both the gold-POS and predicted-POS setup. The differences in standard deviation are unsurprisingly smaller in the predicted POS setup. In general, the rule-based UPD is less sensitive to domain shifts than the data-driven MSD counterpart, confirming earlier findings (Plank and van Noord, 2010).

Table 7 gives the detailed scores per language and domain. From the scores we can see that presidental `bulletin`, `legal` and `weblogs` are amongst the hardest domains to parse. However, the systems often do not agree on which domain is hardest, with the exception of Bulgarian

---

[7]The three in-house annotated datasets are freely available at http://ANONYMIZED

`bulletin`. More importantly, this might even change between gold and predicted POS, highlighting the importance of evaluating systems beyond gold POS. Interestingly, for the Italian data and some of the hardest domains UDP outperforms MSD, confirming that it is a robust baseline.

## 7 Conclusion

We have presented UDP, an unsupervised dependency parser for Universal Dependencies that makes use of personalized PageRank and a small set of head-dependent rules. The parser requires no training data and estimates adpositon direction directly from test data. We achieve competitive performance on all but two UD languages, and even beat a multi-source delexicalized parser (MSD) on Hindi. We evaluated the parser on three POS setups and across domains. Our results show that UDP is less affected by deteriorating POS tags than MSD, and is more resilient to domain changes. Both the parser and the in-domain annotated test sets are freely available.[8]

Further work includes extending the parser to handle multiword expressions, coordination, and proper names. Moreover, our usage of PR could be expanded to directly score the potential dependency edges—e.g., by means of edge reification—instead of words. Finally, we only considered unlabeled attachment, however, our system could easily be augmented with partial edge labeling.

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
