# Peer review of "Parsing for Universal Dependencies without training"

_CoNLL 2016 — decision unknown_

[Official Review · Reviewer 1 · rating 3 · confidence 4]
soundness 4 · originality 2 · clarity 4 · impact 2 · substance 3 · appropriateness 5 · meaningful comparison 5 · replicability 4 · presentation format Poster

This paper describes a new deterministic dependency parsing algorithm and
analyses its behaviour across a range of languages.
The core of the algorithm is a set of rules defining permitted dependencies
based on POS tags.
The algorithm starts by ranking words using a slightly biased PageRank over a
graph with edges defined by the permitted dependencies.
Stepping through the ranking, each word is linked to the closest word that will
maintain a tree and is permitted by the head rules and a directionality
constraint.

Overall, the paper is interesting and clearly presented, though seems to differ
only slightly from Sogaard (2012), "Unsupervised Dependency Parsing without
Training".
I have a few questions and suggestions:

Head Rules (Table 1) - It would be good to have some analysis of these rules in
relation to the corpus.
For example, in section 3.1 the fact that they do not always lead to a
connected graph is mentioned, but not how frequently it occurs, or how large
the components typically are.

I was surprised that head direction was chosen using the test data rather than
training or development data.
Given how fast the decision converges (10-15 sentences), this is not a major
issue, but a surprising choice.

How does tie-breaking for words with the same PageRank score work?
Does it impact performance significantly, or are ties rare enough that it
doesn't have an impact?

The various types of constraints (head rules, directionality, distance) will
lead to upper bounds on possible performance of the system.
It would be informative to include oracle results for each constraint, to show
how much they hurt the maximum possible score.
That would be particularly helpful for guiding future work in terms of where to
try to modify this system.

Minor:

- 4.1, "we obtain [the] rank"

- Table 5 and Table 7 have columns in different orders. I found the Table 7
arrangement clearer.

- 6.1, "isolate the [contribution] of both"

[Official Review · Reviewer 2 · rating 4 · confidence 4]
soundness 4 · originality 4 · clarity 5 · impact 4 · substance 5 · appropriateness 5 · meaningful comparison 5 · replicability 5 · presentation format Oral Presentation

The authors proposed an unsupervised algorithm for Universal Dependencies that
does not require training. The tagging is based on PageRank for the words and a
small amount of hard-coded rules.
The article is well written, very detailed and the intuition behind all prior
information being added to the model is explained clearly.
I think that the contribution is substantial to the field of unsupervised
parsing, and the possibilities for future work presented by the authors give
rise to additional research.

[Official Review · Reviewer 3 · rating 3 · confidence 4]
soundness 4 · originality 2 · clarity 4 · impact 3 · substance 3 · appropriateness 5 · meaningful comparison 3 · replicability 4 · presentation format Poster

This paper presents a way to parse trees (namely the universal dependency
treebanks) by relying only on POS and by using a modified version of the
PageRank to give more way to some meaningful words (as opposed to stop words).

This idea is interesting though very closed to what was done in SÃ¸gaard
(2012)'s paper. The personalization factor giving more weight to the main
predicate is nice but it would have been better to take it to the next level.
As far as I can tell, the personalization is solely used for the main predicate
and its weight of 5 seems arbitrary.

Regarding the evaluation and the detailed analyses, some charts would have been
beneficial, because it is sometimes hard to get the gist out of the tables.
Finally, it would have been interesting to get the scores of the POS tagging in
the prediction mode to be able to see if the degradation in parsing performance
is heavily correlated to the degradation in tagging performance (which is what
we expect).

All in all, the paper is interesting but the increment over the work of
SÃ¸gaard (2012) is small.

Smaller issues:
-------------------

l. 207 : The the main idea -> The main idea